# SpectralNet: Spectral Clustering using Deep Neural Networks

**Uri Shaham**[*][†]**, Kelly Stanton,**[*] **Henry Li**[*]
Yale University
New Haven, CT, USA
{uri.shaham, kelly.stanton, henry.li}@yale.edu

**Boaz Nadler, Ronen Basri**
Weizmann Institute of Science
Rehovot, Israel
{boaz.nadler, ronen.basri}@gmail.com

**Yuval Kluger**
Yale University
New Haven, CT, USA
yuval.kluger@yale.edu

## Abstract

Spectral clustering is a leading and popular technique in unsupervised data analysis. Two of its major limitations are scalability and generalization of the spectral embedding (i.e., out-of-sample-extension). In this paper we introduce a deep learning approach to spectral clustering that overcomes the above shortcomings. Our network, which we call *SpectralNet*, learns a map that embeds input data points into the eigenspace of their associated graph Laplacian matrix and subsequently clusters them. We train SpectralNet using a procedure that involves constrained stochastic optimization. Stochastic optimization allows it to scale to large datasets, while the constraints, which are implemented using a special-purpose output layer, allow us to keep the network output orthogonal. Moreover, the map learned by SpectralNet naturally generalizes the spectral embedding to unseen data points. To further improve the quality of the clustering, we replace the standard pairwise Gaussian affinities with affinities learned from the given unlabeled data using a Siamese network. Additional improvement of the resulting clustering can be achieved by applying the network to code representations produced, e.g., by standard autoencoders. Our end-to-end learning procedure is fully unsupervised. In addition, we apply VC dimension theory to derive a lower bound on the size of SpectralNet. State-of-the-art clustering results are reported on the Reuters dataset. Our implementation is publicly available at https://github.com/kstant0725/SpectralNet.

## 1 Introduction

Discovering clusters in unlabeled data is a task of significant scientific and practical value. With technological progress images, texts, and other types of data are acquired in large numbers. Their labeling, however, is often expensive, tedious, or requires expert knowledge. Clustering techniques provide useful tools to analyze such data and to reveal its underlying structure.

Spectral Clustering (Shi & Malik, 2000; Ng et al., 2002; Von Luxburg, 2007) is a leading and highly popular clustering algorithm. It works by embedding the data in the eigenspace of the Laplacian matrix, derived from the pairwise similarities between data points, and applying $k$-means to this representation to obtain the clusters. Several properties make spectral clustering appealing: First, its embedding optimizes a natural cost function, minimizing pairwise distances between similar data points; moreover, this optimal embedding can be found analytically. Second, spectral clustering variants arise as relaxations of graph balanced-cut problems (Von Luxburg, 2007). Third, spectral clustering was shown to outperform other popular clustering algorithms such as $k$-means

---

[*]Equal contribution.
[†]Also at Final Research, Herzliya, Israel.

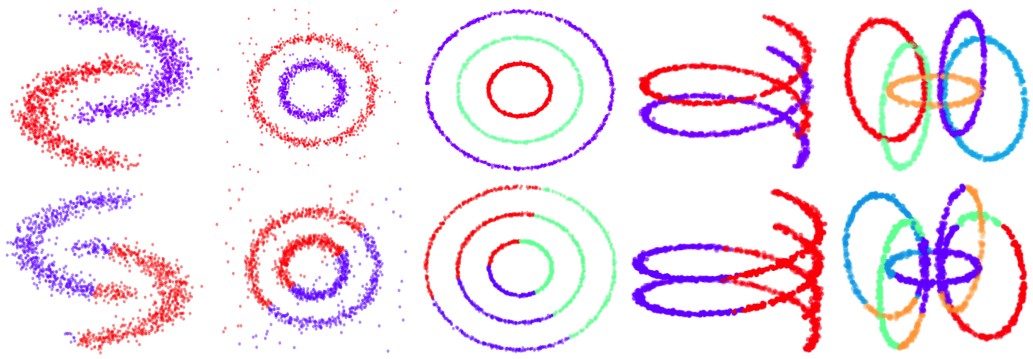

Figure 1: Illustrative 2D and 3D examples showing the results of our SpectralNet clustering (top) compared to typical results obtained with DCN, VaDE, DEPICT and IMSAT (bottom) on simulated datasets in 2D and 3D. Our approach successfully finds these non-convex clusters, whereas the competing algorithms fail on all five examples. (The full set of results for these algorithms is shown in Figure 4 in Appendix A.)

(Von Luxburg, 2007), arguably due to its ability to handle non-convex clusters. Finally, it has a solid probabilistic interpretation, since the Euclidean distance in the embedding space is equal to a diffusion distance, which, informally, measures the time it takes probability mass to transfer between points, via all the other points in the dataset (Nadler et al., 2006; Coifman & Lafon, 2006a).

While spectral embedding of data points can be achieved by a simple eigen-decomposition of their graph Laplacian matrix, with large datasets direct computation of eigenvectors may be prohibitive. Moreover, generalizing a spectral embedding to unseen data points, a task commonly referred to as out-of-sample-extension (OOSE), is a non-trivial task; see, for example, (Belkin et al., 2006; Bengio et al., 2004; Fowlkes et al., 2004; Coifman & Lafon, 2006b).

In this work we introduce SpectralNet, a deep learning approach to spectral clustering, which addresses the scalability and OOSE problems pointed above. Specifically, SpectralNet is trained in a stochastic fashion, which allows it to scale. Moreover, once trained, it provides a function, implemented as a feed-forward network, that maps each input data point to its spectral embedding coordinates. This map can easily be applied to new test data. Unlike optimization of standard deep learning models, SpectralNet is trained using constrained optimization, where the constraint (orthogonality of the net outputs) is enforced by adding a linear layer, whose weights are set by the QR decomposition of its inputs. In addition, as good affinity functions are crucial for the success of spectral clustering, rather than using the common Euclidean distance to compute Gaussian affinity, we show how Siamese networks can be trained from the given *unlabeled* data to learn more informative pairwise distances and consequently significantly improve the quality of the clustering. Further improvement can be achieved if our network is applied to transformed data obtained by an autoencoder (AE). On the theoretical front, we utilize VC-dimension theory to derive a lower bound on the size of neural networks that compute spectral clustering. Our experiments indicate that our network indeed approximates the Laplacian eigenvectors well, allowing the network to cluster challenging non-convex point sets, which recent deep network based methods fail to handle; see examples in Figure 1. Finally, SpetralNet achieves competitive performance on MNIST handwritten digit dataset and state-of-the-art on the Reuters document dataset, whose size makes standard spectral clustering inapplicable.

## 2 RELATED WORK

Recent deep learning approaches to clustering largely attempt to learn a code for the input that is amenable to clustering according to either the $k$-means or mixture of gaussians clustering models. DCN (Yang et al., 2017) directly optimizes a loss composed of a reconstruction term (for the code) and the $k$-means functional. DEC (Xie et al., 2016) iteratively updates a target distribution to sharpen cluster associations. DEPICT (Dizaji et al., 2017) adds a regularization term that prefers balanced clusters. All three methods are pre-trained as autoencoders, while DEPICT also initializes

its target distribution using $k$-means or other standard clustering algorithms. Several other recent approaches rely on a variational autoencoder that utilizes a Gaussian mixture prior, see, for example, VaDE (Zheng et al., 2016) and GMVAE (Dilokthanakul et al., 2016). IMSAT (Hu et al., 2017) is based on data augmentation, where the net is trained to maximize the mutual information between inputs and predicted clusters, while regularizing the net so that the cluster assignment of original data points will be consistent with the assignment of augmented points. Different approaches are proposed by Chen (2015), who uses a deep belief net followed by non-parametric maximum margin clustering (NMMC), and by Yang et al. (2016), who introduce a recurrent-agglomerative framework to image clustering.

While these approaches achieve accurate clustering results on standard datasets (such as the MNIST and Reuters), the use of the $k$-means criterion, as well as the Gaussian mixture prior, seems to introduce an implicit bias towards the formation of clusters with convex shapes. This limitation seems to hold even in code space. This bias is demonstrated in Figure 1(bottom), which shows the failure of several of the above approaches on relatively simple clustering tasks. In contrast, as is indicated in Figure 1(top), our SpectralNet approach appears to be less vulnerable to such bias. The full set of runs can be found in Appendix A.

In the context of spectral clustering, Tian et al. (2014) learn an autoencoder that maps the rows of a graph Laplacian matrix onto the corresponding spectral embedding, and then use $k$-means in code space to cluster the underlying data. Unlike our work, which learns to map an input data point to its spectral embedding, Tian et al.'s network takes as input an *entire row* of the graph Laplacian, and therefore OOSE is impractical, as it requires to compute the affinities of each new data point to all the training data. Also of interest is the kernel spectral method by Alzate & Suykens (2010), which allows for out of sample extension and handles large datasets through smart sampling (but does not use a neural network).

Yi et al. (2016) address the problem of 3D shape segmentation. Their work, which focuses on learning graph convolutions, uses a graph spectral embedding through eigenvector decomposition, which is *not learned*. In addition, we enforce orthogonalization stochastically through a constraint layer, while they attempt to learn orthogonalized functional maps by adding an orthogonalization term to the loss function, which involves non-trivial balancing between two loss components.

Other deep learning works use a spectral approach in the context of *supervised* learning. Law et al. (2017) apply supervised metric learning, showing that their method approximates the eigenvectors of a 0-1 affinity matrix constructed from the true labels. Mishne et al. (2017) trained a network to compute graph Laplacian eigenvectors using supervised regression. Their approach, however, requires the true eigenvectors for training, and hence does not easily scale to large datasets.

Finally, a number of papers showed that stochastic gradient descent can be used effectively to compute the principal components of covariance matrices, see, e.g., (Shamir, 2015) and references therein. The setup in these papers assumes that in each iteration a noisy estimate of the entire input matrix is provided. In contrast, in our work we use in each iteration only a small submatrix of the affinity matrix, corresponding to a small minibatch. In future work, we plan to examine how these algorithms can be adapted to improve the convergence rate of our proposed network.

## 3 SPECTRALNET

In this section we present our proposed approach, describe its key components, and explain its connection to spectral clustering. Consider the following standard clustering setup: Let $\mathcal{X} = \{x_1, \ldots, x_n\} \subseteq \mathbb{R}^d$ denote a collection of *unlabeled* data points drawn from some unknown distribution $\mathcal{D}$; given a target number of clusters $k$ and a distance measure between points, the goal is to learn a similarity measure between points in $\mathcal{X}$ and use it to learn a map that assigns each of $x_1, \ldots, x_n$ to one of $k$ possible clusters, so that similar points tend to be grouped in the same cluster. As in classification tasks we further aim to use the learned map to determine the cluster assignments of new, yet unseen, points drawn from $\mathcal{D}$. Such *out-of-sample-extension* is based solely on the learned map, and requires neither computation of similarities between the new points and the training points nor re-clustering of combined data.

In this work we propose *SpectralNet*, a neural network approach for spectral clustering. Once trained, SpectralNet computes a map $F_\theta : \mathbb{R}^d \to \mathbb{R}^k$ and a cluster assignment function $c : \mathbb{R}^k \to$

$\{1, \ldots, k\}$. It maps each input point $x$ to an output $y = F_\theta(x)$ and provides its cluster assignment $c(y)$. The spectral map $F_\theta$ is implemented using a neural network, and the parameter vector $\theta$ denotes the network weights.

The training of SpectralNet consists of three components: (i) unsupervised learning of an affinity given the input distance measure, via a Siamese network (see Section 3.2); (ii) unsupervised learning of the map $F_\theta$ by optimizing a spectral clustering objective while enforcing orthogonality (see Section 3.1); (iii) learning the cluster assignments, by k-means clustering in the embedded space.

## 3.1 Learning the Spectral Map $F_\theta$

In this section we describe the main learning step in SpectralNet, component (ii) above. To this end, let $w : \mathbb{R}^d \times \mathbb{R}^d \to [0, \infty)$ be a symmetric affinity function, such that $w(x, x')$ expresses the similarity between $x$ and $x'$. Given $w$, we would like points $x, x'$ which are similar to each other (i.e., with large $w(x, x')$) to be embedded close to each other. Hence, we define the loss

$$\mathcal{L}_{\text{SpectralNet}}(\theta) = \mathbb{E}\left[ w(x, x') \|y - y'\|^2 \right], \tag{1}$$

where $y, y' \in \mathbb{R}^k$, the expectation is taken with respect to pairs of i.i.d. elements $(x, x')$ drawn from $\mathcal{D}$, and $\theta$ denotes the parameters of the map $y = F_\theta(x)$. Clearly, the loss $\mathcal{L}_{\text{SpectralNet}}(\theta)$ can be minimized by mapping all points to the same output vector ($F_\theta(x) = y_0$ for all $x$). To prevent this, we require that the outputs will be orthonormal in expectation with respect to $\mathcal{D}$, i.e.,

$$\mathbb{E}\left[ y y^T \right] = I_{k \times k}. \tag{2}$$

As the distribution $\mathcal{D}$ is unknown, we replace the expectations in (1) and (2) by their empirical analogues. Furthermore, we perform the optimization in a stochastic fashion. Specifically, at each iteration we randomly sample a minibatch of $m$ samples, which without loss of generality we denote $x_1, \ldots, x_m \in \mathcal{X}$, and organize them in an $m \times d$ matrix $X$ whose $i$th row contains $x_i^T$. We then minimize the loss

$$L_{\text{SpectralNet}}(\theta) = \frac{1}{m^2} \sum_{i,j=1}^{m} W_{i,j} \|y_i - y_j\|^2, \tag{3}$$

where $y_i = F_\theta(x_i)$ and $W$ is a $m \times m$ matrix such that $W_{i,j} = w(x_i, x_j)$. The analogue of (2) for a small minibatch is

$$\frac{1}{m} Y^T Y = I_{k \times k}, \tag{4}$$

where $Y$ is a $m \times k$ matrix of the outputs whose $i$th row is $y_i^T$.

We implement the map $F_\theta$ as a general neural network whose last layer enforces the orthogonality constraint (4). This layer gets input from $k$ units, and acts as a linear layer with $k$ outputs, where the weights are set to orthogonalize the output $Y$ for the minibatch $X$. Let $\tilde{Y}$ denote the $m \times k$ matrix containing the inputs to this layer for $X$ (i.e., the outputs of $F_\theta$ over the minibatch before orthogonalization). A linear map that orthogonalizes the columns of $\tilde{Y}$ is computed through its QR decomposition. Specifically, for any matrix $A$ such that $A^T A$ is full rank, one can obtain the QR decomposition via the Cholesky decomposition $A^T A = L L^T$, where $L$ is a lower triangular matrix, and then setting $Q = A \left( L^{-1} \right)^T$. This is verified in Appendix B. Therefore, in order to orthogonalize $\tilde{Y}$, the last layer multiplies $\tilde{Y}$ from the right by $\sqrt{m} \left( \tilde{L}^{-1} \right)^T$, where $\tilde{L}$ is obtained from the Cholesky decomposition of $\tilde{Y}^T \tilde{Y}$ and the $\sqrt{m}$ factor is needed to satisfy (4).

We train this spectral map in a coordinate descent fashion, where we alternate between orthogonalization and gradient steps. Each of these steps uses a different minibatch (possibly of different sizes), sampled uniformly from the training set $\mathcal{X}$. In each orthogonalization step we use the QR decomposition to tune the weights of the last layer. In each gradient step we tune the remaining weights using standard backpropagation. Once SpectralNet is trained, all the weights are frozen, including those of the last layer, which simply acts as a linear layer. Finally, to obtain the cluster assignments $c_1, \ldots c_2$, we propagate $x_1, \ldots x_n$ through it to obtain the embeddings $y_1, \ldots, y_n \in \mathbb{R}^k$, and perform $k$-means on them, obtaining $k$ cluster centers, as in standard spectral clustering. These algorithmic steps are summarized below in Algorithm 1 in Sec. 3.3.

**Connection with Spectral Clustering**      The loss (3) can also be written as

$$L_{\text{SpectralNet}}(\theta) = \frac{2}{m^2} \operatorname{trace}\left(Y^T(D - W)Y\right),$$

where $D$ is a $m \times m$ diagonal matrix such that $D_{i,i} = \sum_j W_{i,j}$. The symmetric, positive semi-definite matrix $D - W$ forms the (unnormalized) graph Laplacian of the minibatch $x_1, \ldots, x_m$. For $k = 1$ the loss is minimized when $y$ is the eigenvector of $D - W$ corresponding to the smallest eigenvalue. Similarly, for general $k$, under the constraint (4), the minimum is attained when the column space of $Y$ is the subspace of the $k$ eigenvectors corresponding to the smallest $k$ eigenvalues of $D - W$. Note that this subspace includes the constant vector whose inclusion does not affect the final cluster assignment.

Hence, SpectralNet approximates spectral clustering, where the main differences are that the training is done in a stochastic fashion, and that the orthogonality constraint with respect to the *full* dataset $\mathcal{X}$ holds only approximately. SpectralNet therefore trades accuracy with scalability and generalization ability. Specifically, while its outputs are an approximation of the true eigenvectors, the stochastic training enables its scalability and thus allows one to cluster large datasets that are prohibitive for standard spectral clustering. Moreover, once trained, SpectralNet provides a parametric function whose image for the training points is (approximately) the eigenvectors of the graph Laplacian. This function can now naturally embed new test points, which were not present at training time. Our experiments with the MNIST dataset (Section 5) indicate that the outputs of SpectralNet closely approximate the true eigenvectors.

Finally, as in common spectral clustering applications, cluster assignments are determined by applying $k$-means to the embeddings $y_1, \ldots y_n$. We note that the $k$-means step can be replaced by other clustering algorithms. Our preference to use $k$-means is based on the interpretation (for normalized Laplacian matrices) of the Euclidean distance in the embedding space as diffusion distance in the input space (Nadler et al., 2006; Coifman & Lafon, 2006a).

**Normalized graph Laplacian**      In spectral clustering, the symmetric normalized graph Laplacian $I - D^{-\frac{1}{2}}WD^{-\frac{1}{2}}$ can use as an alternative to the unnormalized Laplacian $D - W$. In order to train SpectralNet with normalized graph Laplacian, the loss function (3) should be replaced by

$$L_{\text{SpectralNet}}(\theta) = \frac{1}{m^2} \sum_{i,j=1}^{m} W_{i,j} \left\| \frac{y_i}{d_i} - \frac{y_j}{d_j} \right\|^2, \tag{5}$$

where $d_i = D_{i,i} = \sum_{j=1}^{m} W_{i,j}$.

**Batch size considerations**      Typically in classification or regression, the loss is a sum over the losses of individual examples. In contrast, SpectralNet loss (3) is summed over pairs of points, and each summand describes relationships between data points. This relation is encoded by the full $n \times n$ affinity matrix $W_{\text{full}}$ (which we never compute explicitly). The minibatch size $m$ should therefore be sufficiently large to capture the structure of the data. For this reason, it is also highly important that minibatches will be sampled at random from the entire dataset at each step, and not be fixed across epochs. When the minibatches are fixed, the knowledge of $W_{\text{full}}$ is reduced to a (possibly permuted) diagonal sequence of $m \times m$ blocks, thus ignoring many of the entries of $W_{\text{full}}$. In addition, while the output layer orthogonalizes $\tilde{Y}$, we do not have any guarantees on how well it orthogonalizes other random minibatches. However, in our experiments we observed that if $m$ is large enough, it approximately orthogonalizes other batches as well, and its weights stabilize as training progresses. Therefore, to train SpectralNet, we use larger minibatches compared to common choices made by practitioners in the context of classification. In our experiments we use minibatches of size 1024 for MNIST and 2048 for Reuters, re-sampled randomly at every step.

### 3.2   Learning affinities using a Siamese network

Choosing a good affinity measure is crucial to the success of spectral clustering. In many applications, practitioners use an affinity measure that is positive for a set of nearest neighbor pairs, combined with a Gaussian kernel with some scale $\sigma > 0$, e.g.,

$$W_{i,j} = \begin{cases} \exp\left(-\frac{\|x_i - x_j\|^2}{2\sigma^2}\right), & x_j \text{ is among the nearest neighbors of } x_i \\ 0, & \text{otherwise,} \end{cases} \tag{6}$$

where one typically symmetrizes $W$, for example, by setting $W_{i,j} \leftarrow (W_{i,j} + W_{j,i})/2$.

Euclidean distance may be overly simplistic measure of similarity; seeking methods that can capture more complex similarity relations might turn out advantageous. Siamese nets (Hadsell et al., 2006; Shaham & Lederman, 2018) are trained to learn affinity relations between data points; we empirically found that the *unsupervised* application of a Siamese net to determine the distances often improves the quality of the clustering.

Siamese nets are typically trained on a collection of similar (positive) and dissimilar (negative) pairs of data points. When labeled data are available, such pairs can be chosen based on label information (i.e., pairs of points with the same label are considered positive, while pairs of points with different labels are considered negative). Here we focus on datasets that are unlabeled. In this case we can learn the affinities directly from Euclidean proximity or from graph distance, e.g., by "labeling" points $x_i, x_j$ positive if $\|x_i - x_j\|$ is small and negative otherwise. In our experiments, we construct positive pairs from the nearest neighbors of each point. Negative pairs are constructed from points with larger distances. This Siamese network, therefore, is trained to learn an adaptive nearest neighbor metric.

A Siamese net maps every data point $x_i$ into an embedding $z_i = G_{\theta_{\text{siamese}}}(x_i)$ in some space. The net is typically trained to minimize contrastive loss, defined as

$$L_{\text{siamese}}(\theta_{\text{siamese}}; x_i, x_j) = \begin{cases} \|z_i - z_j\|^2, & (x_i, x_j) \text{ is a positive pair} \\ \max(c - \|z_i - z_j\|, 0))^2, & (x_i, x_j) \text{ is a negative pair}, \end{cases}$$

where $c$ is a margin (typically set to 1).

Once the Siamese net is trained, we use it to define a batch affinity matrix $W$ for the training of SpectralNet, by replacing the Euclidean distance $\|x_i - x_j\|$ in (6) with $\|z_i - z_j\|$.

Remarkably, despite being trained in an unsupervised fashion on a training set constructed from relatively naive nearest neighbor relations, in Section 5 we show that affinities that use the Siamese distances yield dramatically improved clustering quality over affinities that use Euclidean distances. This implies that unsupervised training of Siamese nets can lead to learning useful and rich affinity relations.

### 3.3 ALGORITHM

Our end-to-end training approach is summarized in Algorithm 1.

**Input**: $\mathcal{X} \subseteq \mathbb{R}^d$, number of clusters $k$, batch size $m$;
**Output**: embeddings $y_1, \ldots, y_n$, $y_i \in \mathbb{R}^k$, cluster assignments $c_1, \ldots c_n$, $c_i \in \{1, \ldots k\}$
Construct a training set of positive and negative pairs for the Siamese network;
Train a Siamese network;
Randomly initialize the network weights $\theta$;
**while** $L_{SpectralNet}(\theta)$ *not converged* **do**
    **Orthogonalization step:**
    Sample a random minibatch $X$ of size $m$;
    Forward propagate $X$ and compute inputs to orthogonalization layer $\tilde{Y}$;
    Compute the Cholesky factorization $LL^T = \tilde{Y}^T \tilde{Y}$;
    Set the weights of the orthogonalization layer to be $\sqrt{m} \left(L^{-1}\right)^T$;
    **Gradient step:**
    Sample a random minibatch $x_1, \ldots, x_m$;
    Compute the $m \times m$ affinity matrix $W$ using the Siamese net;
    Forward propagate $x_1, \ldots, x_m$ to get $y_1, \ldots, y_m$;
    Compute the loss (3) or (5);
    Use the gradient of $L_{\text{SpectralNet}}(\theta)$ to tune all $F_\theta$ weights, except those of the output layer;
**end**
Forward propagate $x_1, \ldots, x_n$ and obtain $F_\theta$ outputs $y_1, \ldots, y_n$;
Run $k$-means on $y_1, \ldots, y_n$ to determine cluster centers;

**Algorithm 1:** SpectralNet training

Once SpectralNet is trained, computing the embeddings of new test points (i.e., out-of-sample-extension) and their cluster assignments is straightforward: we simply propagate each test point $x_i$ through the network $F_\theta$ to obtain their embeddings $y_i$, and assign the point to its nearest centroid, where the centroids were computed using $k$-means on the training data, at the last line of Algorithm 1.

### 3.4 SPECTRAL CLUSTERING IN CODE SPACE

Given a dataset $\mathcal{X}$, one can either apply SpectralNet in the original input space, or in a code space (obtained, for example, by an autoencoder). A code space representation is typically lower dimensional, and often contains less nuisance information (i.e., information on which an appropriate similarity measure should not depend). Following (Yang et al., 2017; Xie et al., 2016; Zheng et al., 2016) and others, we empirically observed that SpectralNet performs best in code space. Unlike these works, which use an autoencoder as an initialization for their clustering networks, we use the code as our data representation and apply SpectralNet directly in that space, (i.e., we do not change the code space while training SpectralNet). In our experiments, we use code spaces obtained from publicly available autoencoders trained by Zheng et al. (2016) on the MNIST and Reuters datasets.

## 4 THEORETICAL ANALYSIS

Our proposed SpectralNet not only determines cluster assignments in training, as clustering algorithms commonly do, but it also produces a map that can generalize to unseen data points at test time. Given a training set with $n$ points, it is thus natural to ask how large should such a network be to represent this spectral map. The theory of VC-dimension can provide useful worst-case bounds for this size.

In this section, we use the VC dimension theory to study the minimal size a neural network should have in order to compute spectral clustering for $k = 2$. Specifically, we consider the class of functions that map all training points to binary values, determined by thresholding at zero the eigenvector of the graph Laplacian with the second smallest eigenvalue. We denote this class of binary classifiers $\mathcal{F}_n^{\text{spectral clustering}}$. Note that with $k = 2$, $k$-means can be replaced by thresholding of the second smallest eigenvector, albeit not necessarily at zero. We are interested in the minimal number of weights and neurons required to allow the net to compute such functions, assuming the affinities decay exponentially with the Euclidean distance. We do so by studying the VC dimension of function classes obtained by performing spectral clustering on $n$ points in arbitrary Euclidean spaces $\mathbb{R}^d$, with $d \geq 3$. We will make no assumption on the underlying distribution of the points.

In the main result of this section, we prove a lower bound on the VC dimension of spectral clustering, which is linear in the number of points $n$. In contrast, the VC dimension of $k$-means, for example, depends solely on the dimension $d$, but not on $n$, hence making $k$-means significantly less expressive than spectral clustering[1]. As a result of our main theorem, we bound from below the number of weights and neurons in any net that is required to compute Laplacian eigenvectors. The reader might find the analysis in this section interesting in its own right.

Our main result shows that for data in $\mathbb{R}^d$ with $d \geq 3$, the VC dimension of $\mathcal{F}_n^{\text{spectral clustering}}$ is linear in the number $n$ of points, making spectral clustering almost as rich as arbitrary clustering of the $n$ points.

**Theorem 4.1.** *VC dim*$(\mathcal{F}_n^{\text{spectral clustering}}) \geq \frac{1}{10}n$.

The formal proof of Theorem 4.1 is deferred to Appendix C. Below we informally sketch its principles. We show that for any integer $n$ (assuming for simplicity that $n$ is divisible by 10), there exists a set of $m = n/10$ points in $\mathbb{R}^d$ that is shattered by $\mathcal{F}_n^{\text{spectral clustering}}$. In particular, we show this for the set of $m$ points placed in a 2-dimensional grid in $\mathbb{R}^d$. We then show that for any arbitrary dichotomy of these $m$ points, we can augment the set of points to a larger set $X$, containing $n = 10m$ points, with a balanced partition of $X$ into two disjoint sets $S$ and $T$ that respects the dichotomy of the

---

[1]For two clusters in $\mathbb{R}^d$, $k$-means clustering partitions the data using a linear separation. It is well known that the VC dimension of the class of linear classifiers in $\mathbb{R}^d$ is $d+1$. Hence, $k$-means can shatter at most $d+1$ points in $\mathbb{R}^d$, regardless of the size $n$ of the dataset.

original $m$ points. The larger set has the special properties: (1) within $S$ (and resp. $T$), there is a path between any two points such that the distances between all pairs of consecutive points along the path are small, and (2) all pairs $(s, t) \in S \times T$ are far apart. We complete the proof by constructing a Gaussian affinity $W$ with a suitable value of $\sigma$ and showing that the minimizer of the spectral clustering loss for $(X, W)$ (i.e., the second eigenvector of the Laplacian), when thresholded at 0, separates $S$ from $T$, and hence respects the original dichotomy.

By connecting Theorem 4.1 with known results regarding the VC dimension of neural nets, see, e.g., (Shalev-Shwartz & Ben-David, 2014), we can bound the size from below (in terms of number of weights and neurons) of any neural net that computes spectral clustering. This is formalized in the following corollary.

**Corollary 4.2.**

1. *For the class of neural nets with $|v|$ sigmoid nodes and $|w|$ weights to represent all functions realizable by spectral clustering (i.e., second eigenvector of the Laplacian, thresholded at 0) on $n$ points, it is necessary to have $|w|^2|v|^2 \geq O(n)$.*

2. *For the class of neural nets with $|w|$ weights from a finite family (e.g., single-precision weights) to represent all functions realizable by spectral clustering, it is necessary to have $|w| \geq O(n)$.*

*Proof.*

1. The VC dimension of the class of neural nets with $|v|$ sigmoid units and $|w|$ weights is at most $O(|w|^2|v|^2)$ (Shalev-Shwartz & Ben-David, 2014, p. 275). Hence, if $|w|^2|v|^2 < O(n)$, such net cannot shatter any collection of points of size $O(n)$. From Theorem 4.1, $\mathcal{F}_n^{\text{spectral clustering}}$ shatters at least $O(n)$ points. Therefore, in order for a class of networks to be able to express any function that can be computed using spectral clustering, it is a necessary (but not sufficient) condition to satisfy $|w|^2|v|^2 \geq O(n)$.

2. The VC dimension of the class of neural nets with $|w|$ weights from a finite family is $O(w)$ (Shalev-Shwartz & Ben-David, 2014, p. 276). The arguments above imply that $|w| \geq O(n)$.

$\square$

Corollary 4.2 implies that in the general case (i.e., without assuming any structure on the $n$ data points), to perform spectral clustering, the size of the net has to grow with $n$. However, when the data has some geometric structure, the net size can be much smaller. Indeed, in a related result, the ability of neural networks to learn eigenvectors of Laplacian matrices was demonstrated both empirically and theoretically by Mishne et al. (2017). They proved that there exist networks which approximate the eigenfunctions of manifold Laplacians arbitrarily well (where the size of the network depends on the desired error and the parameters of the manifold, but not on $n$).

## 5 EXPERIMENTAL RESULTS

### 5.1 EVALUATION METRICS

To numerically evaluate the accuracy of the clustering, we use two commonly used measures, the *unsupervised clustering accuracy* (ACC), and the *normalized mutual information* (NMI). For completeness, we define ACC and NMI below, and refer the reader to (Cai et al., 2011) for more details. For data point $x_i$, let $l_i$ and $c_i$ denote its true label and predicted cluster, respectively. Let $l = (l_1, \ldots l_n)$ and similarly $c = (c_1, \ldots c_n)$.

ACC is defined as

$$\text{ACC}(l, c) = \frac{1}{n} \max_{\pi \in \Pi} \sum_{i=1}^{n} \mathbb{1} \{l_i = \pi(c_i)\},$$

where $\Pi$ is the collection of all permutations of $\{1, \ldots k\}$. The optimal permutation $\pi$ can be computed using the Kuhn-Munkres algorithm (Munkres, 1957).

| Algorithm | ACC (MNIST) | NMI (MNIST) | ACC (Reuters) | NMI (Reuters) |
|---|---|---|---|---|
| $k$-means | .534 | .499 | .533 | .401 |
| Spectral clustering | .717 | .754 | NA | NA |
| DEC | .843[*] | .8[**] | .756[*] | not reported |
| DCN | .83[**] | .81[**] | not reported | not reported |
| VaDE | .9446[†] | not reported | .7938[†] | not reported |
| JULE | not reported | .913[‡] | not reported | not reported |
| DEPICT | .965[††] | .917[††] | not reported | not reported |
| IMSAT | **.984**±**.004**[‡‡] | not reported | .719 [‡‡] | not reported |
| SpectralNet (input space, Euclidean distance) | .622±.008 | .687±.004 | .645±.01 | .444±.01 |
| SpectralNet (input space, Siamese distance) | .826±.03 | .884±.02 | .661± 017 | .381 ± .057 |
| SpectralNet (code space, Euclidean distance) | .800±.003 | .814±.008 | .605±.053 | .401±.061 |
| **SpectralNet (code space, Siamese distance)** | .971±.001 | .924±.001 | **.803**±**.006** | **.532**±**.010** |

Table 1: Performance of various clustering methods on MNIST and Reuters datasets. ([*]) reported in (Xie et al., 2016). ([**]) reported in (Yang et al., 2017), ([†]) reported in (Zheng et al., 2016), ([‡]) reported in (Dizaji et al., 2017), ([††]) reported in (Yang et al., 2016), ([‡‡]) reported in (Hu et al., 2017). The IMSAT result on Reuters was obtained on a subset of 10,000 from the full dataset.

NMI is defined as

$$\text{NMI}(l, c) = \frac{I(l; c)}{\max\{H(l), H(c)\}},$$

where $I(l; c)$ denotes the mutual information between $l$ and $c$, and $H(\cdot)$ denotes their entropy. Both ACC and NMI are in $[0, 1]$, with higher values indicating better correspondence the clusters and the true labels.

## 5.2 CLUSTERING

We compare SpectralNet to several deep learning-based clustering approaches on two real world datasets. In all runs we assume the number of clusters is given (k=10 in MNIST and k=4 in Reuters). As a reference, we also report the performance of $k$-means and (standard) spectral clustering. Specifically, we compare SpectralNet to DEC (Xie et al., 2016), DCN (Yang et al., 2017), VaDE (Zheng et al., 2016), JULE (Yang et al., 2016), DEPICT (Dizaji et al., 2017), and IMSAT (Hu et al., 2017). The results for these six methods are reported in the corresponding papers. Technical details regarding the application of $k$-means and spectral clustering appear in Appendix D.

We considered two variants of Gaussian affinity functions: using Euclidean distances (6), and Siamese distances; the latter case follows Algorithm 1. In all experiments we used the loss (3). In addition, we report results of SpectralNet (and the Siamese net) in both input space and code space. The code spaces are obtained using the publicly available autoencoders which are used to pre-train the weights of VaDE[2], and are 10-dimensional. We refer the reader to Appendix D for technical details about the architectures and training procedures.

### 5.2.1 MNIST

MNIST is a collection of 70,000 $28 \times 28$ gray-scale images of handwritten digits, divided to training (60,000) and test (10,000) sets. To construct positive pairs for the Siamese net, we paired each instance with its two nearest neighbors. An equal number of negative pairs were chosen randomly from non-neighboring points.

Table 1 shows the performance of the various clustering algorithms on the MNIST dataset, using all 70,000 images for training. As can be seen, the performance of SpectralNet is significantly improved when using Siamese distance instead of Euclidean distance, and when the data is represented in code space rather than in pixel space. With these two components, SpectralNet outperforms DEC, DCN, VaDE, DEPICT and JULE, and is competitive with IMSAT.

To evaluate how well the outputs of SpectralNet approximate the true eigenvectors of the graph Laplacian, we compute the Grassmann distance between the subspace of SpectralNet outputs and that of the true eigenvectors. The squared Grassmann distance measures the sum of squared sines

---

[2] https://github.com/slim1017/VaDE/tree/master/pretrain_weights

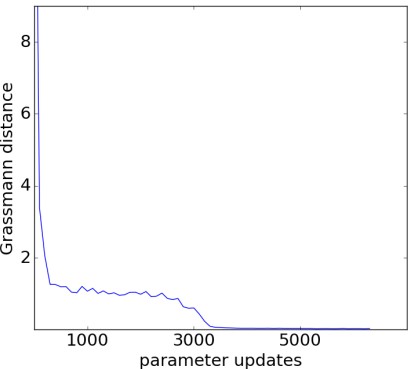

Figure 2: Grassmann distance as a function of iteration update for the MNIST dataset.

of the angles between two $k$-dimensional subspaces; the distance is in $[0, k]$. Figure 2 shows the Grassmann distance on the MNIST dataset as a function of the training time (expressed as number of parameter updates). It can be seen that the distance decreases rapidly at the beginning of training and stabilizes around 0.026 as time progresses.

To check the generalization ability of SpectralNet to new test points, we repeated the experiment, this time training SpectralNet only on the training set, and predicting the labels of the test examples by passing them through the net and associating each test example with the nearest centroid from the $k$-means that were performed on the embedding of the training examples. The accuracy on test examples was .970, implying that SpectralNet generalizes well to unseen test data in this case. We similarly also evaluated the generalization performance of k-means. The accuracy of k-means on the test set is .546 when using the input space and .776 when using the code space, both significantly inferior to SpectralNet.

### 5.2.2 REUTERS

The Reuters dataset is a collection of English news, labeled by category. Like DEC and VaDE, we used the following categories: corporate/industrial, government/social, markets, and economics as labels and discarded all documents with multiple labels. Each article is represented by a tf-idf vector, using the 2000 most frequent words. The dataset contains $n = 685,071$ documents. Performing vanilla spectral clustering on a dataset of this size in a standard way is prohibitive. The AE used to map the data to code space was trained based on a random subset of 10,000 samples from the full dataset. To construct positive pairs for the Siamese net, we randomly sampled 300,000 examples from the entire dataset, and paired each one with a random neighbor from its 3000 nearest neighbors. An equal number of negative pairs was obtained by randomly pairing each point with one of the remaining points.

Table 1 shows the performance of the various algorithms on the Reuters dataset. Overall, we see similar behavior to what we observed on MNIST: SpectralNet outperforms all other methods, and performs best in code space, and using Siamese affinity. Our SpectralNet implementation took less than 20 minutes to learn the spectral map on this dataset, using a GeForce GTX 1080 GPU. For comparison, computing the top four eigenvectors of the Laplacian matrix of the complete data, needed for spectral clustering, took over 100 minutes using ARPACK. Note that both SpectralNet and spectral clustering require pre-computed nearest neighbor graph. Moreover, spectral clustering using the ARPACK eigenvectors failed to produce reasonable clustering. This illustrates the robustness of our method in contrast to the well known instability of spectral clustering to outliers.

To evaluate the generalization ability of SpectralNet, we divided the data randomly to a 90%-10% split, re-trained the Siamese net and SpectralNet on the larger subset, and predicted the labels of the smaller subset. The test accuracy was 0.798, implying that as on MNIST, SpectralNet generalizes well to new examples.

## 6    CONCLUSIONS

We have introduced SpectralNet, a deep learning approach for approximate spectral clustering. The stochastic training of SpectralNet allows us to scale to larger datasets than what vanilla spectral clustering can handle, and the parametric map obtained from the net enables straightforward out of sample extension. In addition, we propose to use unsupervised Siamese networks to compute distances, and empirically show that this results in better performance, comparing to standard Euclidean distances. Further improvement are achieved by applying our network to code representations produced with a standard stacked autoencoder. We present a novel analysis of the VC dimension of spectral clustering, and derive a lower bound on the size of neural nets that compute it. In addition, we report state of the art results on two benchmark datasets, and show that SpectralNet outperforms existing methods when the clusters cannot be contained in non overlapping convex shapes. We believe the integration of spectral clustering with deep learning provides a useful tool for unsupervised deep learning.

## ACKNOWLEDGEMENTS

We thank Raphy Coifman and Sahand Negahban for helpful discussions. R.B is supported in part by the Minerva foundation with funding from the Federal German Ministry for Education and Research. Y.K and B.N are supported by NIH grant 1R01HG008383-01A1.

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

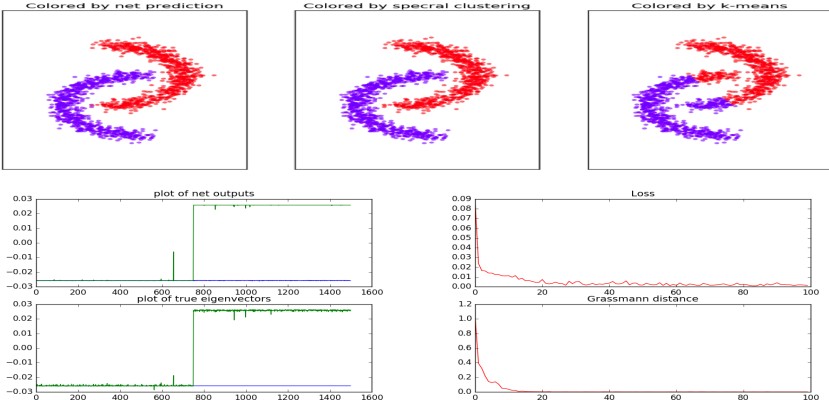

Figure 3: SpectralNet performance on the nested 'C' example. Top row: clustering using SpectralNet (left), spectral clustering (center), and $k$-means (right). Bottom row, left panel: SpectralNet outputs (plotted in blue and green) vs. the true eigenvectors. Bottom row, right panel: loss and Grassmann distance as a function of iteration number; the values on the horizontal axis $\times 100$ are the numbers of the parameter updates.

## A    ILLUSTRATIVE DATASETS

To compare SpectralNet to spectral clustering, we consider a simple dataset of 1500 points in two dimensions, containing two nested 'C'-shaped clusters. We applied spectral clustering to the dataset by computing the eigenvectors of the unnormalized graph Laplacian $L = D - W$ corresponding to the two smallest eigenvalues, and then applying $k$-means (with $k$=2) to these eigenvector embeddings. The affinity matrix $W$ was computed using $W_{i,j} = \exp\left(-\frac{\|x_i - x_j\|^2}{\sigma^2}\right)$, where the scale $\sigma$ was set to be the median distance between a point to its 3rd neighbor – a standard practice in diffusion applications.

Figure 3 shows the clustering of the data obtained by SpectralNet, standard spectral clustering, and $k$-means. It can be seen that both SpectralNet and spectral clustering identify the correct cluster structure, while $k$-means fails to do so. Moreover, despite the stochastic training, the net outputs closely approximate the two true eigenvectors of $W$ with smallest eigenvalues. Indeed the Grassmann distance between the net outputs and the true eigenvectors approaches zero as the loss decreases.

In the next experiment, we trained, DCN, VaDE, DEPICT (using agglomerative clustering initialization) and IMSAT (using adversarial perturbations for data augmentation) on the 2D datasets of Figure 1. The experiments were performed using the code published by the authors of each paper. For each method, we tested various network architectures and hyper-parameter settings. Unfortunately, we were unable to find a setting that will yield an appropriate clustering on any of the datasets for DCN, VaDE and DEPICT. IMSAT worked on two out of the five datasets, however failed to yield an appropriate clustering in fairly simple cases. Plots with typical results of each of the methods on each of the five 2D datasets is shown in Figure 4.

To further investigate why these methods fail, we performed a sequence of experiments with the two nested 'C's data, while changing the distance between the two clusters. The results are shown in Figure 5. We can see that all three methods fail to cluster the points correctly once the clusters cannot be linearly separated.

Interestingly, although the target distribution of DEPICT was initialized with agglomerative clustering, which successfully clusters the nested 'C's, its target distribution becomes corrupted throughout the training, although its loss is considerably reduced, see Figure 6.

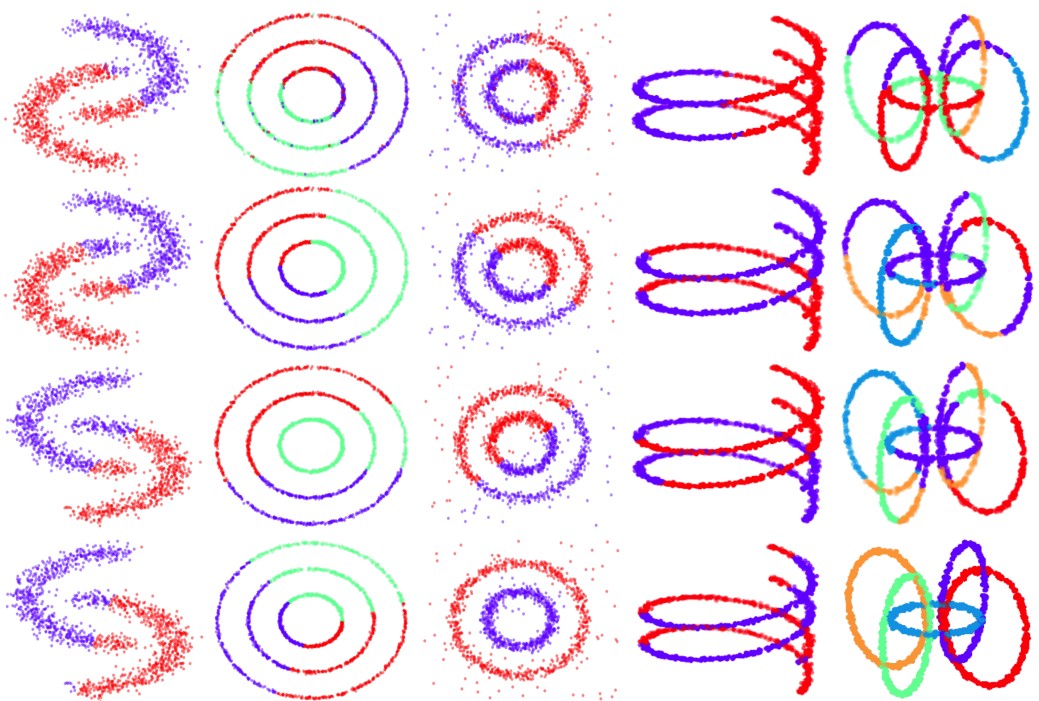

Figure 4: from top to bottom: Results of DCN, VaDE, DEPICT and IMSAT on our illustrative datasets.

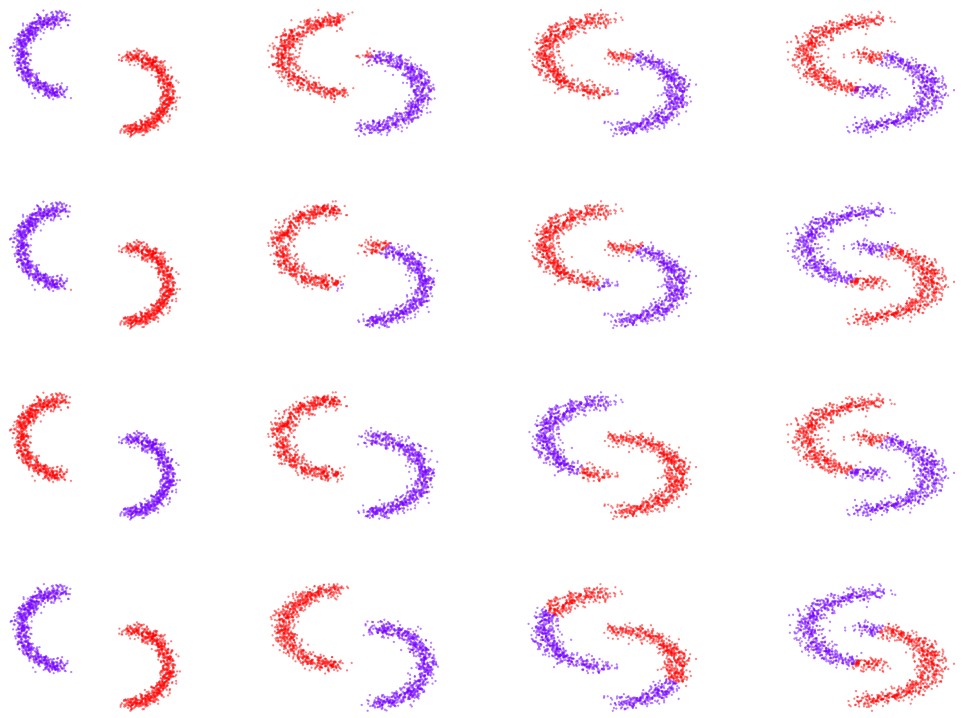

Figure 5: From top: Typical results of DCN, VaDE, DEPICT and IMSAT on the nested 'C's, with several different distances between the two clusters.

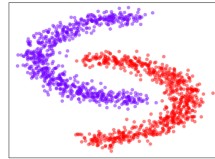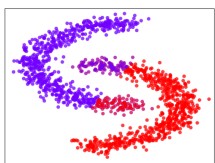

Figure 6: The nested 'C's, colored by DEPICT target distribution. Left: at initialization (with agglomerative clustering initialization). the DEPICT loss at this stage is 9.01. Right: after DEPICT training. The loss is 0.22. Although the loss decreases with training, the target distribution becomes corrupted.

## B    CORRECTNESS OF THE $QR$ DECOMPOSITION

We next verify that the Cholesky decomposition can indeed be used to compute the QR decomposition of a positive definite matrix. First, observe that since $L$ is lower triangular, then so is $L^{-1}$, and $(L^{-1})^T$ is upper triangular. Hence for $i = 1, \ldots m$, the column space of the first $i$ columns of $A$ is the same as the column space of the first $i$ columns of $Q = A(L^{-1})^T$. To show that the columns of $Q$ corresponds to Gram-Schmidt orthogonalization of the columns of $A$, it therefore remains to show that $Q^T Q = I$. Indeed:

$$Q^T Q = L^{-1} A^T A (L^{-1})^T = L^{-1} L L^T (L^{-1})^T = (L^{-1} L)^T = I.$$

## C    SECTION 4 PROOFS

### C.1    PRELIMINARIES

To prove Theorem 4.1, we begin with the following definition and lemmas.

**Definition C.1** ($(\alpha, \beta)$-separated graph). *Let $\alpha > \beta \geq 0$. An $(\alpha, \beta)$-separated graph is $G = (V, W)$, where $V$ has an even number of vertices and has a balanced partition $V = S \cup T$, $|S| = |T|$, and $W$ is an affinity matrix so that:*

- *For any $v_i, v_j \in S$ (resp. $T$), there is a path $v_i = v_{k_1}, v_{k_2}, \ldots, v_{k_l} = v_j \in S$, so that for every two consecutive points $v_{k_l}, v_{k_{l+1}}$ along the path, $W_{k_l, k_{l+1}} \geq \alpha$.*

- *For any $v_i \in S$, $v_j \in T$, $W_{i,j} \leq \beta$.*

**Lemma C.2.** *For any integer $m > 0$ there exists a set $\tilde{X} = \{x_1, \ldots, x_m\} \subseteq \mathbb{R}^d$ $(d \geq 3)$, so that for any binary partition $\tilde{X} = \tilde{S} \cup \tilde{T}$, we can construct a set $X$ of $n = 10m$ points, $\tilde{X} \subset X$, and a balanced binary partition $X = S \cup T$, $|S| = |T|$ of it, such that*

- *$\tilde{S} \subset S$, $\tilde{T} \subset T$*

- *For any $x_i, x_j \in S$ (resp. $T$), there is a path $x_i, x_{k_1}, x_{k_2}, \ldots, x_{k_l}, x_j \in S$, so that for every two consecutive points $x_{k_l}, x_{k_{l+1}}$ along the path, $\|x_{k_l} - x_{k_{l+1}}\| \leq b < 1$ **(property a)**.*

- *For any $x_i \in S$, $x_j \in T$, $\|x_i - x_j\| \geq 1$**(property b)**.*

*Proof.* We will prove this for the case $d = 3$; the proof holds for any $d \geq 3$.

Let $m > 0$ be integer. We choose the set $\tilde{X}$ to lie in a 2-dimensional unit grid inside a square of minimal diameter, which is placed in the $Z = 0$ plane. Each point $x_i$ is at a distance 1 from its neighbors.

Next, given a partition of $x_1, \ldots, x_m$ to two subsets, $\tilde{S}$ and $\tilde{T}$, we will construct a set $X \supset \tilde{X}$ with $n = 10m$ points and a partition $S \cup T$ that satisfy the conditions of the lemma (an illustration can be seen in Figure 7). First, we add points to obtain a balanced partition. We do so by adding $m$ new points $x_{m+1}, \ldots, x_{2m}$, assigning each of them arbitrarily to either $\tilde{S}$ or $\tilde{T}$ until $|\tilde{S}| = |\tilde{T}| = m$. We

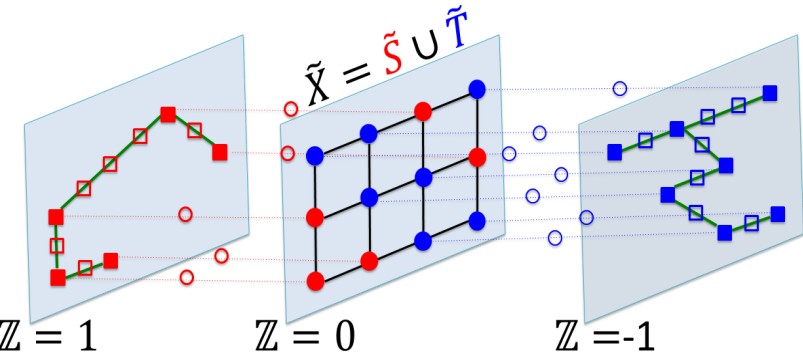

Figure 7: Illustration of the construction of Lemma C.2. We select the set $\tilde{X}$ to lie in a grid in the $Z = 0$ plane. Given an arbitrary dichotomy $\tilde{X} = \tilde{S} \cup \tilde{T}$ (points are marked with filled circles, colored respectively in red and blue), we first add points to make the sets balanced (not shown). Next, we make a copy for $S$ at $Z = 1$ and for $T$ at $Z = -1$ (filled squares). We then add midpoints between each point and its copy (empty circles), and finally add more points along the minimal length spanning tree (empty squares). Together, all the red points form the set $S$; the blue points form the set $T$, and $X = S \cup T$.

place all these points also on grid points in the $Z = 0$ plane so that all $2m$ points lie inside a square of minimal diameter. We further add all the points in $\tilde{S}$ to $S$ and those in $\tilde{T}$ to $T$.

In the next step, we prepare a copy of the $\tilde{S}$-points at $Z = 1$ (with the same $X, Y$ coordinates) and a copy of the $\tilde{T}$-points at $Z = -1$. We denote these copies by $x'_1, ..., x'_{2m}$ and refer to the lifted points at $Z = 1$ by $S'$ and at $Z = -1$ by $T'$. Next, we will add $6m$ more points to make the full set of $n = 10m$ points satisfy properties **a** and **b**. First, we will add the midpoint between every point and its copy, i.e., $x''_i = (x_i + x'_i)/2$. We assign each such midpoint to $S$ (resp. $T$) if it is placed between $x_i \in S$ and $x'_i \in S'$ (resp. $T$ and $T'$). Then we connect the points in $S'$ (resp. $T'$) by a minimal length spanning tree and add $4m$ more points along the edges of these two spanning trees so that the added points are equally spaced along every edge. We assign the new points on the spanning tree of $S'$ to $S$ and of $T'$ to $T$.

We argue that the obtained point set $X$ of size $10m$ satisfies the conditions of the lemma. Clearly, $\tilde{S} \subset S$ and $\tilde{T} \subset T$. To show that property **a** is satisfied, note that the length of each spanning tree cannot exceed $2m$, since the full $2m$ grid points $\tilde{X}$ can be connected with a tree of length $2m - 1$. It is evident therefore that every two points $x_i, x_j \in S$ (resp. $T$) are connected by a path in which the distance between each two consecutive points is strictly less than 1 (property **a**). Property **b** too is satisfied because the grid points in $\tilde{X}$ are at least distance 1 apart; each midpoint $x''_i$ is distance $1/2$ from $x_i$ and $x'_i$ (and they all belong to the same set, either $S$ or $T$), but its distance to the rest of the points in $\tilde{X}$ exceeds 1, and the rest of the points in $S$ (resp. $T$) are on the $Z = 1$ (resp. $Z = 1$) plane, and so they are at least distance 1 away from members of the opposite set which all lie in the $Z \leq 0$ (resp. $Z \geq 0$) half space. $\qquad\square$

**Lemma C.3.** *. Let $f(\cdot)$ be the spectral clustering loss*

$$f(y) = \sum_{i,j} W_{i,j}(y_i - y_j)^2.$$

*Let $G = (X, W)$ be a $(\alpha, \beta)$-separated graph, such that $|X| = n \geq 4$. Let $y^*$ be a minimizer of $f(y)$ w.r.t $W$, subject to $1^T y = 0$, $\|y\| = 1$. Let*

$$\Delta_S = \max\{y^*_i - y^*_j : x_i, x_j \in S\},$$

*and similarly*

$$\Delta_T = \max\{y^*_i - y^*_j : x_i, x_j \in T\}.$$

*Let $\Delta = \max\{\Delta_S, \Delta_T\}$. Then*

$$\frac{\alpha}{\beta}\Delta^2 \leq \frac{n^2}{2}.$$

*Proof.* Without loss of generality, assume that $x_1, \ldots, x_{\frac{n}{2}} \in S$, $x_{\frac{n}{2}+1}, \ldots, x_n \in T$, and that $y_1^* \leq y_2^* \leq \ldots \leq y_{\frac{n}{2}}^*$ and $y_{\frac{n}{2}+1}^* \leq y_{\frac{n}{2}+2}^* \leq \ldots \leq y_n^*$. Also wlog, $\Delta = \Delta_S$. We begin by lower-bounding $f(y^*)$.

$$f(y^*) = \sum_{i,j} W_{i,j}(y_i^* - y_j^*)^2$$
$$\geq \sum_{x_i, x_j \in S} W_{i,j}(y_i^* - y_j^*)^2 + \sum_{x_i, x_j \in T} W_{i,j}(y_i^* - y_j^*)^2.$$

Since $G$ is $(\alpha, \beta)$-separated, there exists a path from $y_1$ to $y_{\frac{n}{2}}$ (and likewise from $y_{\frac{n}{2}+1}$ to $y_n$) where the affinity of every pair of consecutive points exceeds $\alpha$. Denote this path by $\Gamma_S$ (resp. $\Gamma_T$), therefore

$$f(y^*) \geq \alpha \left( \sum_{x_{k_i}, x_{k_{i+1}} \in \Gamma_S} (y_{k_{i+1}}^* - y_{k_i}^*)^2 + \sum_{x_{k_i}, x_{k_{i+1}} \in \Gamma_T} (y_{k_{i+1}}^* - y_{k_i}^*)^2 \right).$$

Note that these are telescopic sums of squares. Clearly, such sum of squares is minimized if all $n/2$ points are ordered and equi-distant, i.e., if we divide a segment of length $\Delta$ into $n/2 - 1$ segments of equal length. Consequently, discarding the second summand,

$$f(y^*) \geq \alpha \left( \frac{n}{2} - 1 \right) \left( \frac{\Delta}{n/2 - 1} \right)^2 = \frac{2\Delta^2 \alpha}{n-2} \geq \frac{2\Delta^2 \alpha}{n},$$

Next, to produce an upper bound, we consider the vector $\bar{y} = \frac{1}{\sqrt{n}}(-1, \ldots, -1, 1, \ldots, 1)$, i.e., $\bar{y}_i = -\frac{1}{\sqrt{n}}$ for $i \leq \frac{n}{2}$, and $\frac{1}{\sqrt{n}}$ otherwise. For this vector,

$$f(\bar{y}) \leq \beta \left( \frac{n}{2} \right)^2 \left( \frac{2}{\sqrt{n}} \right)^2 = n\beta.$$

In summary, we obtain

$$\frac{2\Delta^2 \alpha}{n} \leq f(y^*) \leq f(\bar{y}) \leq n\beta,$$

Hence

$$\frac{\alpha}{\beta} \Delta^2 \leq \frac{n^2}{2}.$$

$\square$

**Lemma C.4.** *Let $y \in \mathbb{R}^n$ be a vector such that $1^T y = 0$, and $\|y\| = 1$. Let $X = S \cup T$, $|S| = |T| = \frac{n}{2}$.*

$$\Delta_S = \max\{y_i - y_j : x_i, x_j \in S\},$$

*and similarly*

$$\Delta_T = \max\{y_i - y_j : x_i, x_j \in T\}.$$

*Let $\Delta = \max\{\Delta_S, \Delta_T\}$. If $\Delta < \frac{1}{\sqrt{2n}}$, then*

$$\max\{y_i : x_i \in S\} < 0 < \min\{y_i : x_i \in T\}.$$

*Proof.* Let

$$m_S = \frac{2}{n} \sum_{x_i \in S} y_i, \quad m_T = \frac{2}{n} \sum_{x_i \in T} y_i.$$

Since $1^T y = 0$, we have $m_S = -m_T$. Without loss of generality, assume that $m_S < 0 < m_T$. For every $y_i$ such that $x_i \in S$,

$$(y_i - m_S)^2 \leq \Delta^2. \tag{7}$$

Similarly, for every $y_i$ such that $x_i \in T$,

$$(y_i + m_S)^2 = (y_i - m_T)^2 \leq \Delta^2.$$

This gives

$$
\begin{aligned}
n\Delta^2 &\geq \sum_{x_i \in S} (y_i - m_S)^2 + \sum_{x_i \in T} (y_i + m_S)^2 \\
&= \sum_{x_i \in S \cup T} y_i^2 - 2m_S \sum_{x_i \in S} y_i + 2m_S \sum_{x_i \in T} y_i + nm_S^2 \\
&= 1 - 2m_S \cdot m_S \frac{n}{2} + 2m_S \cdot -m_S \frac{n}{2} + nm_S^2 \\
&= 1 - nm_S^2,
\end{aligned}
$$

which gives

$$
m_S^2 \geq \frac{1 - n\Delta^2}{n}.
$$

In order to obtain the desired result, i.e., that $\max\{y_i : x_i \in S\} < 0 < \min\{y_i : x_i \in T\}$, it therefore remains to show that for a sufficiently small $\Delta$, by (7), $m_S + \Delta < 0$ (this will also yield $m_T - \Delta > 0$). Hence, we will require

$$
\frac{1 - n\Delta^2}{n} \geq \Delta^2,
$$

which holds for $\Delta < \frac{1}{\sqrt{2n}}$. $\qquad\square$

### C.2 PROOF OF THEOREM 4.1

*Proof.* To determine the VC-dimension of $\mathcal{F}_n^{\text{spectral clustering}}$ we need to show that there exists a set of $m = n/10$ points (assuming for simplicity that $n$ is divisible by 10) that is shattered by spectral clustering. By Lemma C.2, there exists a set of $m$ points $\tilde{X} \subseteq \mathbb{R}^d$ ($d \geq 3$) so that for any dichotomy of $\tilde{X}$ there exists a set $X \supset \tilde{X}$ of $n = 10m$ points, with a balanced partition $X = S \cup T$ that respects the dichotomy of $\tilde{X}$, and whose points satisfy properties **a** and **b** of Lemma C.2 with $0 \leq b < 1$.

Consider next the complete graph $G = (V, W)$ whose vertices $v_i \in V$ correspond to point $x_i$ and the affinity matrix $W$ is set with the standard Gaussian affinity $W_{i,j} = \exp\left(-\frac{\|x_i - x_j\|^2}{2\sigma^2}\right)$, where the value of $\sigma$ will be provided below. It can be readily verified that, due to properties **a** and **b**, $G$ is $(\alpha, \beta)$-separated, where

$$
\alpha = \exp\left(-\frac{b^2}{2\sigma^2}\right), \quad \beta = \exp\left(-\frac{1}{2\sigma^2}\right).
$$

.

Let $y^*$ be the second-smallest eigenvector of the graph Laplacian matrix for $G$, i.e., the minimizer of

$$
f(y) = \sum_{i,j} W_{i,j}(y_i - y_j)^2, \quad \text{s.t.} \quad 1^T y = 0, \; y^T y = 1.
$$

By Lemma C.3, since $G$ is $(\alpha, \beta)$-separated, $\Delta$, i.e, the spread of the entries of $y^*$ for the partition $S \cup T$, should satisfy

$$
\frac{\alpha}{\beta} \Delta^2 \leq \frac{n^2}{2}.
$$

Notice that

$$
\frac{\alpha}{\beta} = \exp\left(\frac{1 - b^2}{2\sigma^2}\right),
$$

allowing us to make $\Delta$ arbitrarily small by pushing the scale $\sigma$ towards $0^3$. In particular, we can set $\sigma$ so as to make $\Delta$ satisfy $\Delta < 1/\sqrt{2n}$. Therefore, by lemma (C.4), thresholding $y^*$ at 0 respects the partition of $X$, and hence also the dichotomy of $\tilde{X}$.

In summary, we have shown that any dichotomy of $\tilde{X}$ can be obtained from a second-smallest eigenvector of some graph Laplacian of $n$ points. Hence the VC dimension of $\mathcal{F}$ is at least $m = n/10$. $\qquad\square$

---

[3]We note that Theorem 4.1 also holds with constant $\sigma$, in which case we can instead uniformly scale the point locations of $\tilde{X}$ and respectively $X$.

|  | Siamese net | SpectralNet |
|---|---|---|
| MNIST | ReLU, size = 1024 | ReLU, size = 1024 |
|  | ReLU, size = 1024 | ReLU, size = 1024 |
|  | ReLU, size = 512 | ReLU, size = 512 |
|  | ReLU, size = 10 | tanh, size = 10 |
|  | - | orthonorm |
| Reuters | ReLU, size = 512 | ReLU, size = 512 |
|  | ReLU, size = 256 | ReLU, size = 256 |
|  | ReLU, size = 128 | tanh, size = 4 |
|  | - | orthonorm |

Table 2: Siamese net and SpectralNet architectures in the MNIST and Reuters experiments.

|  | MNIST Siamese | MNIST SpectralNet | Reuters Siamese | Reuters SpectralNet |
|---|---|---|---|---|
| Batch size | 128 | 1024 | 128 | 2048 |
| Ortho. batch size | - | 1024 | - | 2048 |
| Initial LR | $10^{-3}$ | $10^{-3}$ | $10^{-3}$ | $5 \cdot 10^{-5}$ |
| LR decay | .1 | .1 | .1 | .1 |
| Optimizer | RMSprop | RMSprop | RMSprop | RMSprop |
| Patience epochs | 10 | 10 | 10 | 10 |

Table 3: Additional technical details.

## D  TECHNICAL DETAILS

For $k$-means we used Python's sklearn.cluster; we used the default configuration (in particular, 300 iterations of the algorithm, 10 restarts from different centroid seeds, final results are from the run with the best objective). To perform spectral clustering, we computed an affinity matrix $W$ using (6), with the number of neighbors set to 25 and the scale $\sigma$ set to the median distance from each point to its 25th neighbor. Once $W$ was computed, we took the $k$ eigenvectors of $D - W$ corresponding to the smallest eigenvalues, and then applied $k$-means to that embedding. The $k$-means configuration was as above. In our experiments, the loss (3) was computed with a factor of $\frac{1}{m}$ rather than $\frac{1}{m^2}$, for numerical stability. The architectures of the Siamese net and SpectralNet are described in Table 2. Additional technical details are shown in Table 3.

The learning rate policy for all nets was determined by monitoring the loss on a validation set (a random subset of the training set); once the validation loss did not improve for a specified number of epochs (see *patience epochs* in Table 3), we divided the learning rate by 10 (see *LR decay* in Table 3). Training stopped once the learning rate reached $10^{-8}$. Typical training took about 100 epochs for a Siamese net and less than 20,000 parameter updates for SpectralNet, on both MNIST and Reuters.

In the MNIST experiments, the training set for the Siamese was obtained by pairing each data point with its two nearest neighbors (in Euclidean distance). During the training of the spectral map, we construct the batch affinity matrix $W$ by connecting each point to its nearest two neighbors in the Siamese distance. The scale $\sigma$ in (6) was set to the median of the distances from each point to its nearest neighbor.

In the Reuters experiment, we obtained the training set for the Siamese net by pairing each point from that set to a random point from its 100 nearest neighbors, found by approximate nearest neighbor algorithm[4]. To evaluate the generalization performance, the Siamese nets were trained using training data only. The scale $\sigma$ in (6) was set globally to the median (across all points in the dataset) distance from any point to its 10th neighbor.

Finally, we used the validation loss to determine the hyper-parameters. To demonstrate that indeed the validation loss is correlated to clustering accuracy, we conducted a series of experiments with the MNIST dataset, where we varied the net architectures and learning rate policies; the Siamese net and Gaussian scale parameter $\sigma$ were held fixed throughout all experiments. In each experiment, we measured the loss on a validation set and the clustering accuracy (over the entire data). The correlation between loss and accuracy across these experiments was -0.771. This implies that hyper-

---

[4]https://github.com/spotify/annoy

parameter setting for the spectral map learning can be chosen based on the validation loss, and a setup that yields a smaller validation loss should be preferred. We remark that we also use the convergence of the validation loss to determine our learning rate schedule and stopping criterion.

