# OpenReview forum: "SpectralNet: Spectral Clustering using Deep Neural Networks"
_ICLR.cc/2018/Conference — Accept (Poster)_

### Official Review · AnonReviewer2 · 2017-11-27
**SpectralNet: Spectral Clustering using Deep Neural Networks**

**Rating:** 6
**Confidence:** 3

**Review:**

The authors study deep neural networks for spectral clustering in combination with stochastic optimization for large datasets. They apply VC theory to find a lower bound on the size of the network.

Overall it is an interesting study, though the connections with the existing literature could be strengthened:

- The out-of-sample extension aspects and scalability is stressed in the abstract and introduction to motivate the work.
On the other hand in Table 1 there is only compared with methods that do not possess these properties.
In the literature also kernel spectral clustering has been proposed, possessing out-of-sample properties
and applicable to large data sets, see

``Multiway Spectral Clustering with Out-of-Sample Extensions through Weighted Kernel PCA, IEEE Transactions on Pattern Analysis and Machine Intelligence, vol. 32, no. 2, pp. 335-347, 2010

Sparse Kernel Spectral Clustering Models for Large-Scale Data Analysis, Neurocomputing, vol. 74, no. 9, pp. 1382-1390, 2011

The latter also discussed incomplete Cholesky decomposition which seems related to section 3.1 on p.4.

- related to the neural networks aspects, it would be good to comment on the reproducability of the results with respect to the training results (local minima) and the model selection aspects. How is the number of clusters and number of neurons selected?

---

> ### Author Response · Authors · 2018-01-04
> **Our comments on your review**
>
> Thank you very much for your review!
>
> 1.	Indeed, DEC, DCN, DEPICT, and JULE do not report details on their generalization performance. k-means results on the MNIST test set are now obtained (please see point 5 in our response to reviewer 3). Moreover, our test set performance (reported in the last paragraphs of sections 5.2.1 and 5.2.2) can and should be compared to our training set performance (reported in Table 1). As can be seen, the two are very close, which implies that SpectralNet generalizes well on MNIST and Reuters.
> 2.	The works of Alzate and Suykens mentioned by the reviewer are indeed impressive and very elegant and we thank the reviewer for bringing them to our attention. This work handles large training data by applying smart subsampling. Unlike their approach, our method uses the entire training data. Unfortunately, their work was not tested on standard benchmarks and we were unable to retrieve their code.
> 3.	The reproducibility of our results can be inferred from the standard deviations reported in Table 1. Indeed, our results on MNIST and Reuters are highly reproducible. To verify this further, we repeated our experiments 100 times and obtained very similar results.
> 4.	Specifying an appropriate number of clusters is a general challenge in clustering. In this work we assume that this number is known ahead of time, as pointed out in the first paragraph of Section 3, as well as the input line of Algorithm 1. In our experiments on MNIST and Reuters we simply use the true number or distinct labels (10 for MNIST, 4 for Reuters).
> 5.	Finally, regarding model selection and hyperparameter setting, in order to evaluate the connection between SpectralNet loss and clustering accuracy, we conducted a series of experiments, where we varied the net architectures and and learning rate policies; the Siamese net and Gaussian scale parameter \sigma were held fixed throughout all experiments. In each experiment, we measured the loss on a validation set and the clustering accuracy (over the entire data). The correlation between loss and accuracy across these experiments was -0.771. This implies that hyperparameter setting for the spectral map learning can be chosen based on the validation loss, and a setup that yields a smaller validation loss should be preferred. We remark that we also use the convergence of the validation loss to determine our learning rate schedule and stopping criterion.

---

### Official Review · AnonReviewer3 · 2017-11-27
**Review of SpectralNet**

**Rating:** 4
**Confidence:** 4

**Review:**

PAPER SUMMARY

This paper aims to address two limitations of spectral clustering: its scalability to large datasets and its generalizability to new samples. The proposed solution is based on designing a neural network called SpectralNet that maps the input data to the eigenspace of the graph Laplacian and finds an orthogonal basis for this eigenspace. The network is trained by alternating between orthogonalization and gradient descent steps, where scalability is achieved by using a stochastic optimization scheme that instead of computing an eigendecomposition of the entire data (as in vanilla spectral clustering) uses a Cholesky decomposition of the mini batch to orthogonalize the output. The method can also handle out-of-sample data by applying the learned embedding function to new data. Experiments on the MNIST handwritten digit database and the Reuters document database demonstrate the effectiveness of the proposed SpectralNet.

COMMENTS

1) I find that the output layer (i.e. the orthogonalization layer) is not well-justified. In principle, different batches require different weights on the output layer. Although the authors observe empirically that orthogonalization weights are roughly shared across different batches, the paper lacks a convincing argument for why this can happen. Moreover, it is not clear why an output layer designed to orthogonalized batches from the training set would also orthogonalize batches from the test set?

2) One claimed contribution of this work is that it extends spectral clustering to large scale data. However, the paper could have commented more on what makes spectral clustering not scalable, and how the method in this paper addresses that. The authors did mention that spectral clustering requires computing eigenvectors for large matrices, which is prohibitive. However, this argument is not entirely true, as eigen-decomposition for large sparse matrices can be carried out efficiently by tools such as ARPACK. On the other hand, computing the nearest neighbor affinity or Gaussian affinity is N^2 complexity, which could be the bottleneck of computation for spectral clustering on large scale data. But this issue can be addressed using approximate nearest neighbors obtained, e.g., via hashing. Overall, the paper compares only to vanilla spectral clustering, which is not representative of the state of the art. The paper should do an analysis of the computational complexity of the proposed method and compare it to the computational complexity of both vanilla as well as scalable spectral clustering methods to demonstrate that the proposed approach is more scalable than the state of the art.

3)  Continuing with the point above, an experimental comparison with prior work on large scale spectral clustering (see, e.g. [a] and the references therein) is missing. In particular, the result of spectral clustering on the Reuters database is not reported, but one could use other scalable versions of spectral clustering as a baseline.

4)  Another benefit of the proposed method is that it can handle out-of-sample data. However, the evaluation of such benefit in experiments is rather limited. In reporting the performance on out-of-sample data, there is no other baseline to compare with. One can at least compare with the following baseline: apply k-means to the training data in input space, and classify each test data to the nearest centroid.

5) The reason for using an autoencoder to extract features is unclear. In subspace clustering, it has been observed that features extracted from a scattering transform network [b] can significantly improve clustering performance, see e.g. [c] where all methods have >85% accuracy on MNIST. The methods in [c] are also tested on larger datasets.

[a] Choromanska, et. al., Fast Spectral Clustering via the Nystrom Method, International conference on algorithmic learning theory, 2013

[b] Bruna, Mallat, Invariant Scattering Convolution Networks, arXiv 2012

[c] You, et. al., Oracle Based Active Set Algorithm for Scalable Elastic Net Subspace Clustering, CVPR 2016

---

> ### Author Response · Authors · 2018-01-04
> **Our comments on your review**
>
> Thank you very much for your review!
>
> 1.	The weights of the output layer define a linear transformation that orthogonalizes its input batches. When the batches are small, we agree with the reviewer that different transformations are likely to be needed for different batches. However, when a batch is sufficiently large (as discussed in the last paragraph of section 3.1) and sampled iid from the data distribution, the linear transformation that orthogonalizes it is expected to approximately orthogonalize other (sufficiently large) batches of points sampled in a similar fashion. Indeed, we empirically found that for batch sizes of 2048 on Reuters and 1024 on MNIST, the weights of the output layer also (approximately) orthogonalize the entire dataset.
> 2.	Indeed, there are ways to increase the scalability of vanilla spectral clustering. Methods like ARPACK and PROPACK are very efficient for eigendecomposition of sparse matrices with a rapidly decaying spectrum (which is also the typical case for spectral clustering as well). Following the reviewer’s recommendation, we applied ARPACK to our affinity matrix on the Reuters dataset (n=685,071) using the sparse Gaussian affinities obtained by kNN search with k=3000 neighbors per point (a similar setting to SpectralNet; the number of neighbors is adapted proportionally to the number of neighbors SpectralNet uses in each batch). While SpectralNet takes less than 20 minutes to converge on this dataset (using the same affinity matrix), ARPACK needed 110.4 minutes to obtain the first four eigenvectors of the affinity matrix. We therefore see that SpectralNet scales well compared to ARPACK.
> Please note that both SpectralNet and spectral clustering require pre-computed nearest neighbor graph. In our Reuters experiments we indeed used an approximate nearest neighbor search, which took 20 minutes to run.
> 3.	We performed extensive experiments of spectral clustering using ARPACK on Reuters, using various scales and numbers of neighbors. Unfortunately, we were not able to achieve a reasonable accuracy. We conjecture that this is due of the well-known sensitivity of spectral clustering to noisy data and outliers. SpectralNet, on the other hand, appears to be more robust, due to its stochastic training. This is actually an ongoing research we are currently pursuing.
> 4.	We followed the procedure proposed by the reviewer to evaluate the generalization performance of k-means on MNIST. The accuracy of the test set is .546 when using the input space and .776 when using the code space. Both these results are inferior to SpectralNet performance on this dataset. Moreover, we do compare our performance to a baseline - we actually want the performance on the test data to be similar to the performance on the training data. This is indeed the case in our experiments on both the MNIST and Reuters, as also appears in the manuscript (see the last paragraphs of sections 5.2.2 and 5.2.1, which should be compared to the results in Table 1).
> 5.	Performing the learning task in feature spaces is a standard practice machine learning in general and deep learning in particular. Autoencoders are often used in deep clustering; see for example DCN, Vade, and DEC. Moreover, to be sure that our results were not obtained merely due to a better feature space, we even do not use our own autoencoder; rather, we use the one made publicly available by the authors of VaDE. Finally, SpectralNet can also be applied to the features of a scattering transform, as well as or to any other representation.

---

> > ### Comment · AnonReviewer3 · 2018-01-05
> > **Response to author's rebuttal**
> >
> > I would like to thank the author for addressing our comments.
> >
> > - I find that it is still unclear why there is an orthogonalization layer that can approximately orthogonalize every batch of data. The authors made the argument that, as data is drawn from a certain distribution, it is expected that if two large enough batches are sampled from this distribution they tend to share orthogonalization matrix. However, it is easy to construct counter-examples for this argument. Consider two matrices Y1 and Y2 where Y2 is a permutation of the columns of Y1, i.e. Y2 = Y1*P. They have the same probability of being sampled from any given distribution, but the matrices that orthogonalize them, say R1 and R2, have the relation that R2 = P*R1, are very different. This is our main concern in suggesting a weak rejection as technically the algorithm is unjustified (and incorrect if I may say).
> >
> > - We were suggesting that the paper includes a discussion of prior works on scalable spectral clustering (for which we did not see a response in the rebuttal) as we believe that the reader can better understand the "scalability" of the proposed method in context. We also suggested to include experimental results for scalable spectral clustering methods, although they may not be as good and as fast as the proposed method, so that we can get a better sense of the advantage of the proposed method. Overall, we would also suggest the authors to incorporate their responses in 2, 3, 4 to the paper.

---

> > > ### Author Response · Authors · 2018-01-05
> > > **Response to the reviewer's response**
> > >
> > > We thank the reviewer for clarifying their concern. It seems like the reviewer mistakenly confuses rows with columns. As we explain in detail below, we respectfully maintain that our algorithm is correct, and in particular that permuting the points does not change the orthogonalization layer.
> > >
> > > Using our notation, a minibatch {x_1,...x_m} is arranged in an m x k matrix whose *rows* are the x_i's. For each x_i, the network then produces a non-orthogonalized output \tilde y_i (producing an m x k matrix \tilde Y, whose rows are the \tilde y_i's), and then \tilde y_i is fed to the orthogonalization layer where it is linearly transformed by a k x k matrix \tilde L = \sqrt{m} (L^{-T}) which is supposed to orthogonalize \tilde Y (by multiplying \tilde Y from the right), so that Y = \tilde Y \tilde L is orthogonal (i.e., Y^TY = (1/m)I).
> > >
> > > Suppose now that we obtain a new minimatch that contains the exact same points {x_1,...x_m} but permuted, so X' = PX where P is an m x m permutation matrix -- note that in our notation P permutes the *rows* (i.e. the points), not the columns (the features). This will permute the rows of \tilde Y, and subsequently also the rows of Y, but the result Y' = PY will remain orthogonal, since Y'^TY' = (PY)^T(PY) = Y^T P^T P Y = Y^TY = (1/m)I, where this identity holds since P^T = inverse(P).
> > >
> > > Consequently, as long that each minibatch faithfully represents the data distribution the orthogonalization layer should, at convergence, (roughly) orthogonalize all the minibatches. Our experiments indeed demonstrate that spectral net convergences close to the correct eigenvectors (as can be seen in figure 2, where for MNIST the Grassman distance converges to a low number of 0.026).
> > >
> > > We hope this explanation clarifies our algorithm.
> > >
> > > We also thank you for the other comments and will certainly modify our paper according to our response as requested.

---

### Official Review · AnonReviewer1 · 2017-11-29
**SpectralNet: Spectral Clustering using Deep Neural Networks**

**Rating:** 7
**Confidence:** 5

**Review:**

Brief Summary:
The paper introduces a deep learning based approach that approximates spectral clustering. The basic idea is to train a neural network to map a representation of input in the eigenspace where k-means clustering can be performed as usual. The method is more scalable than the vanilla spectral clustering algorithm and it can also be used to cluster a new incoming data point without redoing the whole spectral clustering procedure. The authors have proved worst case lower bounds on the size of neural network required to perform the task using VC dimension theory. Experiments on MNIST and Reuters dataset show state of the art performance (On Reuter's there is a significant performance improvement under one measure).

Main Contributions:
Introduced SpectralNet - a neural network that maps input points to their embeddings in the eigenspace
Used constraint optimization (using Cholesky decomposition) to train the final layer of neural network to make sure that the output "eigenvectors" are orthonormal
Solves the problem of scalability by using stochastic optimization (basically using mini-batches to train neural network)
Solves the problem of generalization to new data points as the neural network can be used to directly compute the embedding for the incoming data point in the eigenspace
Proved a lower bound for VC dimension of spectral clustering (linear in n as opposed to linear in input dimension d for k-means, which explains the expressive power of spectral clustering)
Derived a worst-case lower bound for the size of neural network that is needed to realize the given objective
Experimented by using Gaussian kernel similarity and similarity learned using a Siamese neural network (trained in an unsupervised way) on both input space and code space (auto-encoder representation)

Overall:
The paper is very clearly written. The idea is simple yet clever. Incorporating the ortho-normalization constraint in the final layer of the neural network is interesting.
The VC dimension based result are interesting but useless as the authors themselves argue that in practical cases the size of neural network required will be much less than the worst case lower bound proved in the paper.
The experiments demonstrate the effectiveness of the proposed approach.
The unsupervised training of Siamese is based on code k-nearest neighbor approach to get positive and negative examples. It is not clear why the learned matrix should outperform Gaussian kernel, but the experiments show that it does.

---

> ### Author Response · Authors · 2018-01-04
> **Our comments on your review**
>
> Thank you very much for your review!
>
> The Siamese net is trained in an unsupervised fashion on pairs which are constructed based on Euclidean nearest neighbor relations. Yet, it learns distances that yield a significant improvement in the clustering performance comparing to the performance using Euclidean distances. We find this empirical observation quite remarkable. To this end, we have several conjectures about the mathematical reason behind this behavior, for example, the ability of this training procedure to exploit local characteristics. Understanding this further is the topic of ongoing work.

---

### Public Comment · (anonymous) · 2017-11-11
**Request for citation**

I believe that you should also cite “Learning Discrete Representations via Information Maximizing Self-Augmented Training” (ICML 2017) http://proceedings.mlr.press/v70/hu17b.html.
This paper is closely related to your work and is also about unsupervised clustering using deep neural networks.
As far as I know, the proposed method, IMSAT, is the current state-of-the-art method in deep clustering (November 2017).
Could you compare your results against their result?

---

> ### Author Response · Authors · 2017-11-11
> **response**
>
> Thank you very much for bringing this work to our attention. We will examine it thoroughly.

---

### Author Response · Authors · 2018-01-04
**Rebuttal - part II**

Reviewer #2:
1.	Indeed, DEC, DCN, DEPICT, and JULE do not report details on their generalization performance. k-means results on the MNIST test set are now obtained (see point 5 above). Moreover, our test set performance (reported in the last paragraphs of sections 5.2.1 and 5.2.2) can and should be compared to our training set performance (reported in Table 1). As can be seen, the two are very close, which implies that SpectralNet generalizes well on MNIST and Reuters.
2.	The works of Alzate and Suykens mentioned by the reviewer are indeed impressive and very elegant and we thank the reviewer for bringing them to our attention. This work handles large training data by applying smart subsampling. Unlike their approach, our method uses the entire training data. Unfortunately, their work was not tested on standard benchmarks and we were unable to retrieve their code.
3.	The reproducibility of our results can be inferred from the standard deviations reported in Table 1. Indeed, our results on MNIST and Reuters are highly reproducible. To verify this further, we repeated our experiments 100 times and obtained very similar results.
4.	Specifying an appropriate number of clusters is a general challenge in clustering. In this work we assume that this number is known ahead of time, as pointed out in the first paragraph of Section 3, as well as the input line of Algorithm 1. In our experiments on MNIST and Reuters we simply use the true number or distinct labels (10 for MNIST, 4 for Reuters).
5.	Finally, regarding model selection and hyperparameter setting, in order to evaluate the connection between SpectralNet loss and clustering accuracy, we conducted a series of experiments, where we varied the net architectures and and learning rate policies; the Siamese net and Gaussian scale parameter \sigma were held fixed throughout all experiments. In each experiment, we measured the loss on a validation set and the clustering accuracy (over the entire data). The correlation between loss and accuracy in these experiments was -0.771. This implies that hyperparameter setting for the spectral map learning can be chosen based on the validation loss, and a setup that yields a smaller validation loss should be preferred. We remark that we also use the convergence of the validation loss to determine our learning rate schedule and stopping criterion.

We thank the reviewers for the thoughtful comments, which we will address in the final version. We hope that at this point the reviewers will be willing to reconsider (in a somewhat positive manner) their rating for this work, which we believe to be novel and important from both algorithmic, theoretical and performance perspectives, and also as a deep learning tool for more general eigendecomposition and manifold learning problems, which may possibly be more stable than current approaches that rely on direct eigendecomposition.

---

### Author Response · Authors · 2018-01-04
**Rebuttal - part I**

We thank the reviewers for their helpful comments.
Below we address the issues pointed out by each reviewer.

Reviewer #1:
1.	The Siamese net is trained in an unsupervised fashion on pairs which are constructed based on Euclidean nearest neighbor relations. Yet, it learns distances that yield a significant improvement in the clustering performance comparing to the performance using Euclidean distances. We find this empirical observation quite remarkable. To this end, we have several conjectures about the mathematical reason behind this behavior, for example, the ability of this training procedure to exploit local characteristics. Understanding this further is the topic of ongoing work.

Reviewer #3
1.	The weights of the output layer define a linear transformation that orthogonalizes its input batches. When the batches are small, we agree with the reviewer that different transformations are likely to be needed for different batches. However, when a batch is sufficiently large (as discussed in the last paragraph of section 3.1) and sampled iid from the data distribution, the linear transformation that orthogonalizes it is expected to approximately orthogonalize other (sufficiently large) batches of points sampled in a similar fashion. Indeed, we empirically found that for batch sizes of 2048 on Reuters and 1024 on MNIST, the weights of the output layer also (approximately) orthogonalize the entire dataset.
2.	Indeed, there are ways to increase the scalability of vanilla spectral clustering. Methods like ARPACK and PROPACK are very efficient for eigendecomposition of sparse matrices with a rapidly decaying spectrum (which is also the typical case for spectral clustering as well). Following the reviewer’s recommendation, we applied ARPACK to our affinity matrix on the Reuters dataset (n=685,071) using the sparse Gaussian affinities obtained by kNN search with k=3000 neighbors per point (a similar setting to SpectralNet; the number of neighbors is adapted proportionally to the number of neighbors SpectralNet uses in each batch). While SpectralNet takes less than 20 minutes to converge on this dataset (using the same affinity matrix), ARPACK needed 110.4 minutes to obtain the first four eigenvectors of the affinity matrix. We therefore see that SpectralNet scales well compared to ARPACK.
Please note that both SpectralNet and spectral clustering require pre-computed nearest neighbor graph. In our Reuters experiments we indeed used an approximate nearest neighbor search, which took 20 minutes to run.
3.	We performed extensive experiments of spectral clustering using ARPACK on Reuters, using various scales and numbers of neighbors. Unfortunately, we were not able to achieve a reasonable accuracy. We conjecture that this is due of the well-known sensitivity of spectral clustering to noisy data and outliers. SpectralNet, on the other hand, appears to be more robust, due to its stochastic training. This is actually an ongoing research we are currently pursuing.
4.	We followed the procedure proposed by the reviewer to evaluate the generalization performance of k-means on MNIST. The accuracy of the test set is .546 when using the input space and .776 when using the code space. Both these results are inferior to SpectralNet performance on this dataset. Moreover, we do compare our performance to a baseline - we actually want the performance on the test data to be similar to the performance on the training data. This is indeed the case in our experiments on both the MNIST and Reuters, as also appears in the manuscript (see the last paragraphs of sections 5.2.2 and 5.2.1, which should be compared to the results in Table 1).
5.	Performing the learning task in feature spaces is a standard practice machine learning in general and deep learning in particular. Autoencoders are often used in deep clustering; see for example DCN, Vade, and DEC. Moreover, to be sure that our results were not obtained merely due to a better feature space, we even do not use our own autoencoder; rather, we use the one made publicly available by the authors of VaDE. Finally, SpectralNet can also be applied to the features of a scattering transform, as well as or to any other representation.

---

### Public Comment · (anonymous) · 2018-04-24
**missing citation**

A (non-deep) neural network implementation of spectral clustering had been proposed in

Ratle, Weston, Miller, Large-Scale Clustering through Functional Embedding, ECML 2008.

Though it is late for modifications, I think this work should be acknowledged as it is very similar.

---

### Decision · Program_Chairs · 2018-01-29
**ICLR 2018 Conference Acceptance Decision**

**Decision:**

Accept (Poster)

**Comment:**

The paper proposes interesting  deep learning based spectral clustering techniques. The use of functional embeddings for enabling spectral clustering to have an out-of-sample extension has of course been explored earlier (e.g., see Manifold Regularization work of Belkin et al, JMLR 2006). For polynomials or kernel-based spectral clustering, the orthogonality of the outputs can be exactly handled via a generalized eigenvector problem, while here the arguments are statistically flavored and not made very clear in the original draft. Some crucial comparisons, e.g., against large-scale versions of vanilla spectral clustering and against other methods that generalize to new samples is missing or not thorough enough. See reviews for more precise description of issues. As such the paper will benefit from a revision.